# Comparison of the kinetics and magnitude of antibody responses to different SARS-CoV-2 proteins in Sinopharm/BBIBP-CorV vaccinees following the BNT162b2 booster or natural infection

**Chandima Jeewandara[1], Inoka Sepali Aberathna[1], Shashika Dayarathna[1], Thashmi Nimasha[1], Thushali Ranasinghe[1], Jeewantha Jayamali[1], Achala Kamaladasa[1], Maneshka Karunanada[1], Lahiru Perera[1], Graham S. Ogg[2], Gathsaurie Neelika Malavige[1,2]***

1 Allergy Immunology and Cell Biology Unit, Department of Immunology and Molecular Medicine, University of Sri Jayewardenepura, Nugegoda, Sri Lanka, 2 MRC Human Immunology Unit, MRC Weatherall Institute of Molecular Medicine, University of Oxford, Oxford, United Kingdom

* gathsaurie.malavige@ndm.ox.ac.uk

## Abstract

The kinetics and magnitude of antibody responses to different proteins of the SARS-CoV-2 virus in Sinopharm/BBIBP-CorV vaccinees has not been previously studied. Therefore, we investigated antibody responses to different SARS-CoV-2 proteins at 2 weeks, 3 months, and 6 months post-second dose in previously infected (n = 20) and uninfected (n = 20) Sinopharm/BBIBP-CorV vaccinees. The IgG antibodies to the S, S1 and S2 and N were several folds higher in those who had natural infection compared to uninfected individuals at all time points. We then compared the persistence of antibody responses and effect of natural omicron infection or BNT162b2 booster in Sinopharm/BBIBP-CorV vaccinees. We measured the total antibodies to the RBD, ACE2 blocking antibodies and antibody responses to different SARS-CoV-2 proteins in Sinopharm vaccinees at 7 months post second dose, including those who remained uninfected and not boosted (n = 21), or those who had previous infection and who did not obtain the booster (n = 17), those who were not infected, but who received a BNT162b2 booster (n = 30), or those who did not receive the booster but were infected with omicron (n = 29). At 7 months post second dose uninfected (no booster) had the lowest antibody levels to the RBD, while omicron infected vaccinees showed significantly higher anti-RBD antibody levels (p = 0.04) than vaccinees who received the booster. Only 3/21 cohort A (14.3%) had ACE2 blocking antibodies, while higher frequencies were observed in naturally infected individuals (100%), those who received the booster (18/21, 85.7%), and omicron infected individuals (100%). Pre-vaccination, naturally infected had the highest antibody levels to the N protein. These data suggest that those previously infected Sinopharm/BBIBP-CorV vaccinees have a robust antibody response, 7 months post vaccination, while vaccinees who were naturally infected with omicron had a similar

**Data Availability Statement:** All relevant data are within the paper and its Supporting Information files.

**Funding:** We are grateful to the funding by the Allergy Immunology and Cell Biology Unit of University of Sri Jayewardenepura (CJ); Sri Lanka Covid 19 Emergency Response and Health Systems Preparedness Project (ERHSP) of Ministry of Health Sri Lanka funded by World Bank (GNM) and the UK Medical Research Council (GSO). The funders had no role in study design, data collection and analysis, decision to publish, or preparation of the manuscript.

**Competing interests:** The authors have declared that no competing interests exist.

immune response to those who received the booster. It will be important to investigate implications for subsequent clinical protection.

## Introduction

Although the SARS-CoV-2 virus continues to evolve and give rise to new variants [1], the global hospitalization rates and number of deaths are currently declining [2]. By end of May 2022, 60% of the global population had been fully vaccinated, with 25.2% receiving the booster [3]. In contrast, in the African continent, only 17% of the population was fully vaccinated and with 1.8% receiving the booster [3]. While COVID-19 vaccines are likely to have played an important role in the decline in global mortality rates and hospitalizations, natural infection may have also contributed to this decline. Among vaccinees, people above 60 years with comorbid conditions were reportedly had higher natural infection and hospitalization rates i.e. about 15 times and 10 times respectively due to rapid decay of antibodies [4]. Although the African continent has very low vaccination rates, the mortality rates have remained lower than other continents, during the two years of the COVID-19 pandemic [2]. However, the reported mortality rates are likely to underestimate the true mortality rates in some regions [5], and many countries in sub-Saharan Africa have reported high excess mortality rates [6].

Due to the declining immune responses following two doses of many of the COVID-19 vaccines, a third dose/booster dose was recommended to all adults [7, 8]. The mRNA-1273 (Moderna) and BNT162b2 (Pfizer) vaccines were the two main vaccines used as the booster dose in many countries irrespective of the type of the primary vaccine [9]. Sri Lanka used several types of vaccines as the primary vaccines for prevention of COVID-19, with Sinopharm/BBIBP-CorV being the most widely used vaccine, with 12 million (70.6%) individuals receiving this vaccine by end of December 2021 [10]. Although the BNT162b2 was offered as a booster dose to Sri Lankan adults in a step-wise manner since November 2021, only 18% had received the booster dose by the end of 2021 [3], when the omicron variant started to rapidly spread in the community being detected in Sri Lanka on the 24th November 2021 [11]. Sinopharm/BBIBP-CorV vaccine was found to be less immunogenic than the mRNA-1273,AZD1222 and Sputnik V, 3 months post second dose, in a head-to-head comparison in the Sri Lankan population [12]. Only 18% of Sri Lankans had received the booster dose when the omicron variant was rapidly transmitted in the community and many individuals who had received the Sinopharm/BBIBP-CorV, were naturally infected with the different omicron variants (BA.1, BA.1.1 and BA.2). However, the mortality rates due to the spread of predominantly the BA.2 omicron variant in Sri Lanka during January and February 2022 in Sri Lanka [11], was associated with lower mortality rates than many European countries and North America [2]. In fact, the mortality rates in Sri Lanka during the height of the BA.2 wave was 0.85/ million individuals compared to 4.03/million individuals in Europe and 5.4/million individuals in North America [2]. This difference is unlikely to be due to under reporting of COVID-19 deaths in Sri Lanka as the excess mortality rates in Sri Lanka were found to be less than the excess mortality rates reported in Europe and North America [6]. Therefore, it is intriguing to consider the reasons for lower mortality rates due to omicron BA.2 in Sri Lanka, which predominantly used a vaccine of lower immunogenicity than Europe and North America.

The Sinopharm/BBIBP-CorV vaccine is an inactivated, whole virus vaccine [13]. Therefore, individuals who receive this vaccine are likely to develop antibody and T cell responses to the spike protein and other structural proteins such as the nucleocapsid, membrane and envelope

proteins. Although neutralizing antibodies directed at the receptor binding domain (RBD) of the spike protein have been shown to associate with protection [14], it was recently shown that spike independent, nucleocapsid vaccine induced protective immunity in animal models [15]. Therefore, it would be important to assess the kinetics of immune responses to different SARS-CoV-2 proteins and their association with protection. In this study, we initially evaluated the kinetics and the magnitude of antibodies to the different SARS-CoV-2 virus proteins in Sinopharm/BBIBP-CorV vaccinees and naturally infected individuals. As only 18% of Sri Lankans had received the BNT162b2 booster when there was rapid transmission of the omicron variants in the community, we carried out a prospective study to investigate the antibody responses to the RBD, ACE2 blocking antibodies and antibodies to different SARS-CoV-2 proteins in Sinopharm/BBIBP-CorV vaccinees, who remained uninfected, or who received the BNT162b2 booster or who were naturally infected with an omicron variant.

## Materials and methods

### Ethics approval

Informed written consent was obtained from all participants. Ethical approval for the study was obtained by the Ethics Review Committee of the University of Sri Jayewardenepura (COVID 01/21).

### Participants for comparison of kinetics and the magnitude of IgG responses to the Sinopharm/BBIBP-CorV vaccine in those with and without natural infection

The kinetics and magnitude of IgG antibody responses against the five proteins following the Sinopharm/BBIBP-CorV was compared with the antibody responses to the vaccine who had natural infection with the virus prior to vaccination using the Peggy Sue platform. Infection status of these individuals was determined by the presence of SARS-CoV-2 RBD specific antibodies in the pre-vaccination serum sample in these individuals using the Wantai SARS-CoV-2 total antibody ELISA (Beijing Wantai Biological Pharmacy Enterprise, China). Accordingly, those who were seropositive prior to vaccination were considered as been naturally infected with the SARS-CoV-2 virus, while those who were seronegative were considered to be uninfected. IgG antibody responses in individuals who were uninfected at the time of receiving the first dose of the vaccine (n = 20), were compared with those who had past-natural infection at the time of receiving the vaccine (n = 20). The IgG antibody responses of these two cohorts were compared at 2 weeks, 3 months, and 6 months post-second dose of the Sinopharm/BBIBP-CorV vaccine (S1 Fig).

### Participants for comparison of magnitude and breadth of SARS-COV-2 antibody responses following natural infection and following the BNT162b2 booster

To investigate the differences in SARS-CoV-2 specific total antibodies to the receptor binding domain (RBD) of the virus following natural infection in comparison to a BNT162b2 booster, we recruited the following groups of individuals. These individuals were recruited between the third week of January to mid-February 2022 (S2 Fig).

A. Sinopharm/BBIBP-CorV who did not have a booster and who were not infected 7 months post second dose: n = 21. Those who were shown to be uninfected pre-vaccination and who did not test positive or the SARS-CoV-2 virus or show symptoms suggestive of

COVID-19 during the 7 months since obtaining the first dose, were considered as being uninfected.

B. Sinopharm/BBIBP-CorV who had previous infection (alpha or delta) and who did not obtain the booster, 7 months post second dose: n = 17. Those who were found to be infected (seropositive) at the time of recruitment and had tested positive during the months of May to July in Sri Lanka, were considered to have been infected with alpha or delta, as they were the only circulating SARS-CoV-2 variants in Sri Lanka during those months.

C. Sinopharm/BBIBP-CorV who were not infected but who received a BNT162b2 booster, 7 months post second dose (1 month post booster): n = 30. Those who were shown to be uninfected pre-vaccination and who did not test positive or the SARS-CoV-2 virus or show symptoms suggestive of COVID-19 during the 7 months since obtaining the first dose, were considered as being uninfected.

D. Sinopharm/BBIBP-CorV who did not receive the booster, but were infected with either BA.1, BA.1.1 or BA.2, 2 weeks post infection (7 months post second dose): n = 29. These were shown to be uninfected prior to vaccination (seronegative) and remained uninfected since obtaining the vaccine for 7 months. During the month of January 2022, they tested positive by realtime qPCR for SARS-CoV-2 infection in our laboratory and their samples were sequenced to identify the variant.

Blood samples were obtained from this cohort at 6 months post second dose of Sinopharm/BBIBP-CorV and 2 weeks after the booster or 2 weeks post infection (cohort C and D).

## Genomic sequencing using Oxford Nanopore (ONT) platform

RT-PCR of nasopharyngeal swabs were carried out on samples using TaqPath COVID-19 CE-IVD RT-PCR kit (Thermo Fisher Scientific, USA) according to the manufacturer's instructions. Nanopore sequencing was carried out as previously described [16]. Briefly, the extracted RNA of 56/190 samples were subjected to nanopore sequencing according to the manufacture's instruction using the SQK-RBK110.96 rapid barcoding kit (ONT, Oxford, UK). 1200 bp tiled PCR amplicons were generated with midnight primers as described in Freed et al., 2020 [17]. All the thermal cycling steps were carried out in a QuantStudio™ 5 Real-Time PCR Instrument (Applied biosystems, Singapore). Barcodes were attached to resulting amplicons and pooled together before the clean-up step. Finally, 800ng of library was loaded into R9.4.1 flow cell and sequenced on the Oxford Nanopore Minion Mk1C. The run was stopped once desired number of reads (~15,000 reads per sample) were achieved.

## Detection of total antibodies to the RBD of the SARS-CoV-2 virus

The presence of total antibodies (IgM, IgG or IgA) to the RBD of the SARS-CoV-2 virus was assessed by using the Wantai SARS-CoV-2 total antibody ELISA (Beijing Wantai Biological Pharmacy Enterprise, China), which was previously found to have a specificity of 100%[18]. Assays were done according to the manufacturer's instructions. The antibody index, which is used as an indirect indicator of the antibody titre was calculated by dividing the absorbance of each sample by the cut-off value.

## Surrogate neutralizing antibody test (sVNT) to detect ACE2 receptor blocking antibodies

The surrogate virus neutralization test (sVNT) (Genscript Biotech, USA) was used to assess ACE2 blocking antibodies as previously described, according to the manufacturer's

instructions. The ACE2 blocking antibody levels were expressed % of inhibition of binding [19, 20] and this test has been used to assess the presence of neutralizing antibodies in several studies previously [21–23]. An inhibition percentage ≥ 25% in a sample was considered as positive for ACE2 blocking antibodies as previously described by us in the Sri Lankan population [19].

### Detection of IgG antibodies to different SARS-CoV-2 proteins

IgG antibodies to the SARS-CoV-2 proteins spike (S), S1 subunit, S2 subunit, nucleocapsid (N) and RBD was performed using the SARS-CoV-2 Multi-Antigen Serology Module (SA-001, ProteinSimple, USA) in the Peggy Sue platform. The serum samples were heat inactivated and diluted with the serum diluent at 1:5. The serum samples were used as primary antibodies against in the Multi-Antigen Mix with S (170 KDa), S1 (98KDa), S2 (69KDa), N (57KDa) and RBD (47 KDa) antigens. However, as the optimization for IgG for RBD was not satisfactory, IgG was only assessed for the remaining four proteins. Anti-Human IgG -HRP Antibody was used as the secondary antibody and fluorescent was detected using luminol peroxide (Anti-Human IgG detection module (DM-005, ProteinSimple, USA). The assays were performed using 12-230KDa separation module (SM-S001, ProteinSimple, USA). Each cycle contained total of 12 capillaries allocating individual capillary for each sample. A biotinylated ladder and a positive control were run in 2 separate capillaries. Human anti-His primary antibody was used in place of donor serum in the positive control. Peak areas were calculated for each SARS-CoV2 antigen in individual serum samples using "Compass for SW". An example of aresponse to the 5 SARS-CoV-2 proteins in an uninfected individual who was vaccinated with Sinopharm/BBIBP-CorV 2 weeks, 3 months and at 6 months post the second dose is shown in S3 Fig.

### Statistical analysis

GraphPad Prism version 8.3 was used for statistical analysis. The differences in antibody responses (total antibodies, ACE2- blocking antibodies and IgG to different SARS-CoV-2 proteins) between the different cohorts was analyzed using the Mann-Whitney test (two-tailed). To analyze the differences in the changes in the levels if IgG to the five different SARS-CoV-2 proteins, Friedman test was used (two-tailed). Corrections for multiple comparisons were done using Holm-Sidak method and the statistically significant value was set at 0.05 (alpha), which was used for the comparisons between the IgG antibodies to different SARS-CoV-2 proteins in infected and uninfected individuals.

## Results

### Magnitude and kinetics of antibody responses to the different SARS-CoV-2 proteins infected and uninfected individuals who received the Sinopharm/BBIBP-CorV

We assessed the changes in the antibody responses to the four different SARS-CoV-2 proteins at 2 weeks, 3 months and 6 months post second dose of the vaccine, in those who were naturally infected (n = 20, Fig 1A) and naïve individuals (n = 20, Fig 1B). The IgG levels to S and S2 remained the same in those who were infected prior to vaccination with a slight and significant reduction in antibodies to S1 and N over time (Fig 1A). In contrast, the antibody responses to all the different proteins significantly reduced over time in uninfected individuals (Fig 1B). There was a significant difference of IgG responses to different proteins at 2 weeks (p = 0.02), 3 months (p = 0.008), but not at 6 months (p = 0.06), with responses lowest for N<S1<S2<S

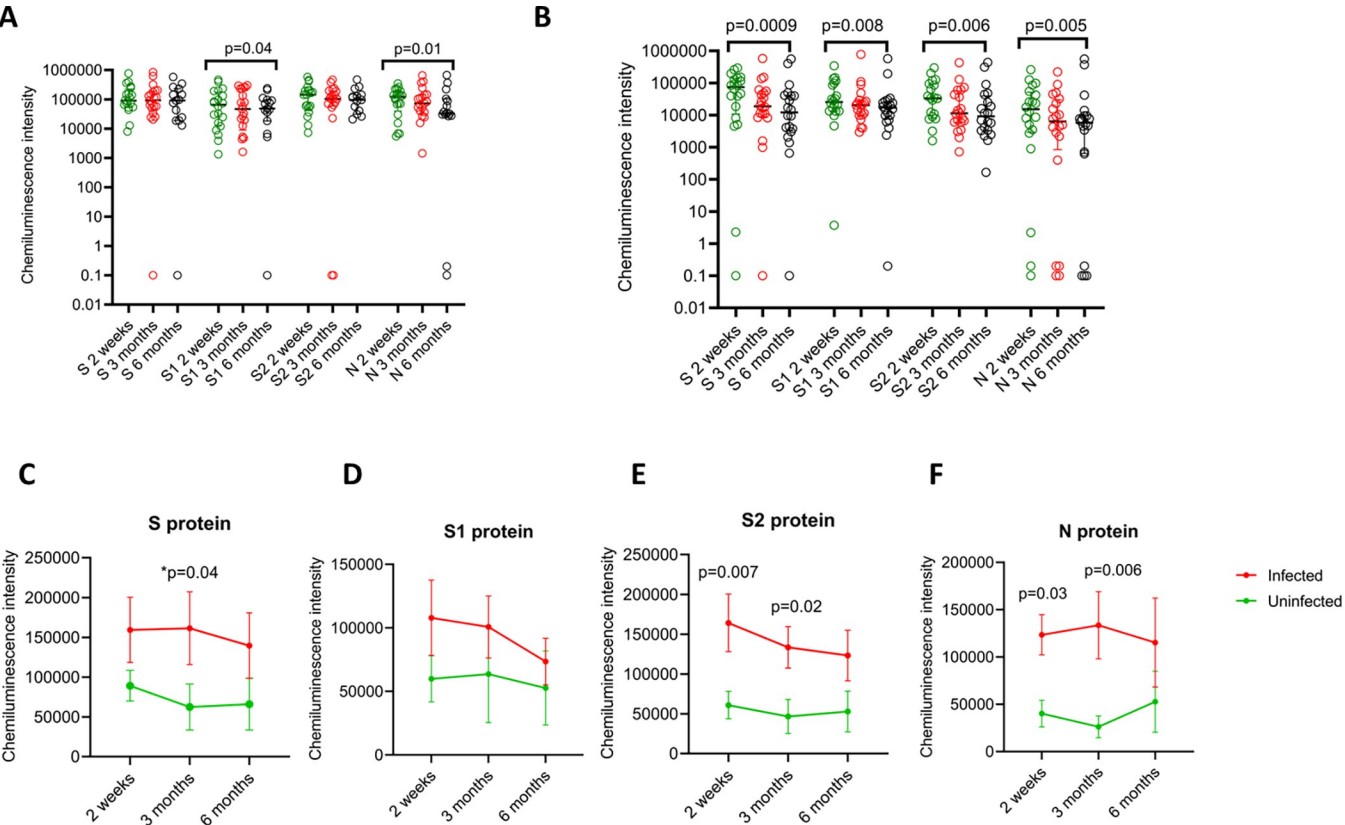

**Fig 1. Kinetics of IgG antibody responses to different SARS-CoV-2 proteins in previously infected and uninfected individuals who received the Sinopharm/BBIBP-CorV vaccine.** IgG antibodies to SARS-CoV-2 spike (S), S1 subunit (S1), S2 subunit (S2) and nucleocapsid (N) at 2 weeks, 3 months and 6 months post second dose, were measured in those who were naturally infected (n = 20) and naïve individuals (n = 20) using the Peggy Sue platform. The IgG responses the different proteins were compared over time in previously infected (A) and uninfected (B) individuals. The changes in responses to S (C), S1 (D), S2 (E) and N (F) was also compared in infected and uninfected individuals over time. The Friedman test was used to compare the IgG levels between different time points for different proteins in infected and uninfected individuals. The Holm-Sidak method was used to compare the differences in IgG levels for different proteins and different time points in those who were infected and uninfected. All tests were two-tailed.

in uninfected individuals. In infected individuals, a significant difference of IgG responses to different proteins was only seen at 3 months (p = 0.04), while the lowest responses were to S1<S, N and S2. The IgG antibodies to the S, S1 and S2 and N were several folds higher in those who had natural infection compared to uninfected individuals at all time points (Fig 1C–1F). This difference was most significant for IgG antibodies specific for the N protein.

## Differences in SARS-CoV-2 RBD specific total antibody responses in Sinopharm/BBIBP-CorV vaccinees who received the BNT162b2 booster or who were naturally infected with omicron

We assessed the antibody responses to the SARS-CoV-2 RBD, in our cohort of individuals (A, B,C and D cohorts) who were uninfected and received the Sinopharm/BBIBP-CorV vaccine 7 months post vaccination (n = 21, group A), infected individuals who received the vaccine (n = 17, group B), uninfected individuals two weeks following the BNT162b2 booster (n = 30, group C), and those who were infected with omicron (BA.1, BA.1.1 and BA.2), 2 weeks after infection (n = 29, group D). Of the 29 individuals who were infected with an omicron variant (based on the absence of the L452R mutation in the PCR), sequencing was carried out in 27 individuals. Of these 27 individuals 5/27 had BA.1, 5/27 had BA.1.1 and 17/27 BA.2. The

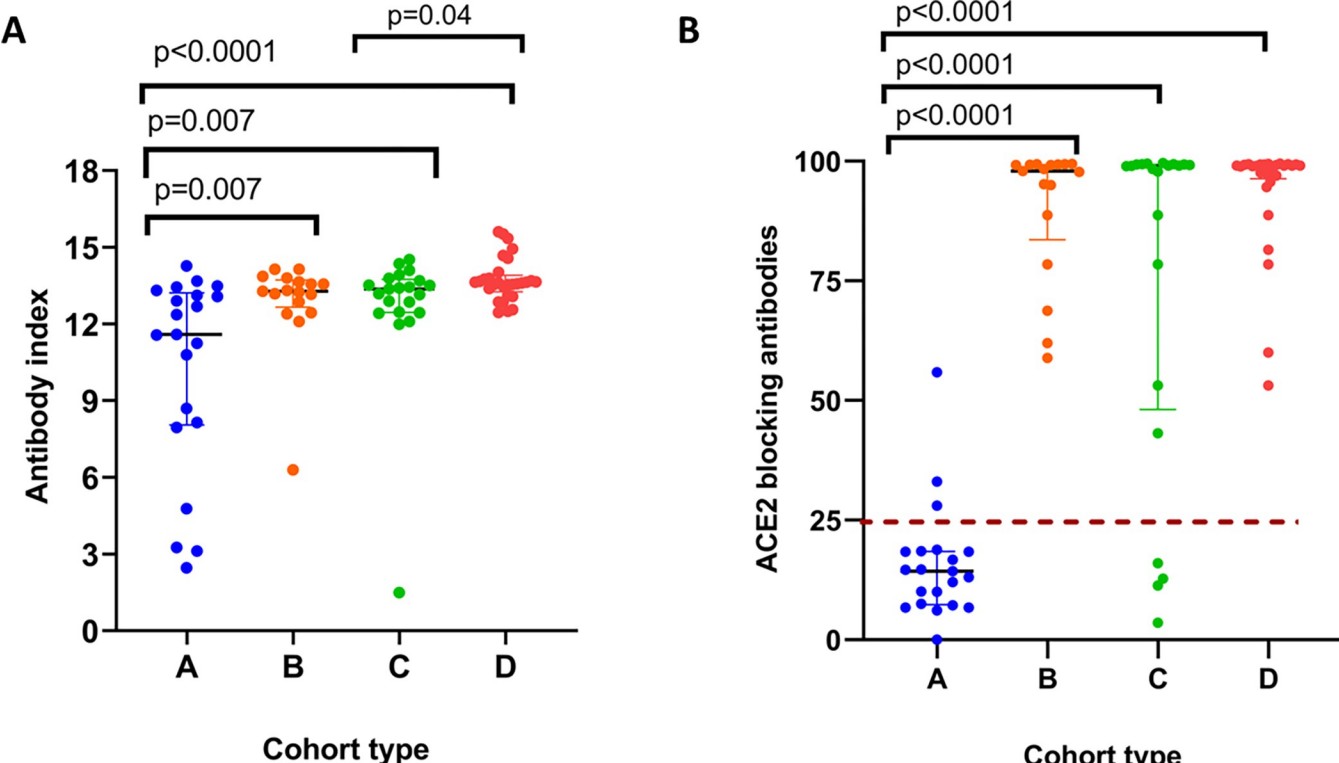

**Fig 2. Antibodies to the SARS-COV-2 in Sinopharm/ BBIBP-CorV vaccinees who either received the BNT162b2 booster or were infected.** The total antibodies to the RBD (A) or ACE2 blocking antibodies (B) were measured by ELISA in Sinopharm vaccinees at 7 months post second dose, who remained uninfected and un-boosted (cohort A: n = 21), or those who had previous infection and who did not obtain the booster (cohort B: n = 17), or those who were not infected but who received a BNT162b2 booster (cohort C: n = 30) or those who did not receive the booster, but were infected with omicron (cohort D: n = 29). The cut-off value for the positive threshold for the surrogate neutralizing assay, measuring the ACE2 blocking antibodies is indicated in a brown dotted line. The Mann-Whitney test was used to compare the IgG levels between different cohorts. All tests were two-tailed.

remaining 2 individuals were also considered to have been infected with omicron variant, as it was the only SARS-CoV-2 variant found in the Colombo district, during the study period (S4 Fig) [24]. The total antibody responses to the RBD of SARS-CoV-2 were significantly higher in the B, C and D cohorts compared to the A cohort (Fig 2A). The total antibodies to the RBD were significantly higher (p = 0.04) in cohort D, compared to cohort C. There was no difference in the antibody levels to the RBD between B, and D cohorts.

## Differences in ACE2 blocking antibody responses Sinopharm/BBIBP-CorV vaccinees who received the BNT162b2 booster or who were naturally infected with omicron

We assessed the ACE2 blocking antibodies in the above cohort of individuals (cohort A to D) using the sVNT assay. The ACE2 blocking antibodies were significantly higher in B, C and D cohorts compared to the A cohort (Fig 2B). However, again there was no difference in ACE2 blocking antibodies between B, C and D cohorts. Of cohort A, only 3/21 (14.3%) individuals had a positive response for ACE2 blocking antibodies. While all individuals (100%) in cohort B had a positive response to the ACE2 blocking antibodies, 18/21 (85.7%) in cohort C and 29/29 (100%) individuals in cohort D, had ACE2 blocking antibodies above the positive cut of threshold.

### Differences in antibody responses the five SARS-CoV-2 proteins in Sinopharm/BBIBP-CorV vaccinees who received the BNT162b2 booster or who were naturally infected with omicron

We assessed the antibody responses to the SARS-CoV-2 RBD, in our cohort of individuals (A, B,C and D cohorts) who were uninfected and received the Sinopharm/BBIBP-CorV vaccine 7 months post vaccination (n = 17), infected individuals who received the vaccine (n = 13), uninfected individuals two weeks following the BNT162b2 booster (n = 13) and those who were infected with omicron (BA.1, BA.1.1 and BA.2), 2 weeks after infection (n = 17) (Fig 3). Those who were not naturally infected before vaccination had significantly lower IgG antibody levels to the spike protein, S1 and S2 than cohorts B, C and D (Fig 3A to 3C). The highest responses to the N protein were seen in those who were infected prior to receiving the Sinopharm/BBIBP-CorV and these individuals had significantly higher responses to the N protein than those who received the BNT162b2 booster and those who were naturally infected with omicron (Fig 3D).

## Discussion

In this study we show that those who were previously infected with the SARS-CoV-2 had significantly higher antibody responses to different SARS-CoV-2 proteins than uninfected individuals and these antibody levels remained high for up to 6 months from vaccination. At 7 months post second dose the vaccinees who remained uninfected had the lowest antibody levels to the RBD, while uninfected individuals who were subsequently infected with omicron showed significantly higher anti-RBD antibody levels than the Sinopharm/BBIBP-CorV vaccinees who received the BNT162b2 booster. As expected, the ACE2 blocking antibodies were lowest in Sinopharm/BBIBP-CorV vaccinees, who remained uninfected, with only 14.3% having a positive antibody response above the cut-off threshold. Although the ACE2 blocking antibody levels were similar in previously infected Sinopharm/BBIBP-CorV vaccinees who did not receive the booster (B), the uninfected vaccinees who received the BNT162b2 booster (C) and Sinopharm/BBIBP-CorV vaccinees, who were infected with omicron (D), all those in the naturally infected cohorts had ACE2 blocking antibodies above the cut-off threshold, compared to 85.7% of those who received the booster. Overall, these data suggest that vaccination along with natural infection is likely to induce more robust immune responses than heterologous single boost vaccine combinations, although this data should be further validated in larger cohorts.

Those who received any of the mRNA vaccines (BNT162b2 or mRNA-1273) and those who received the AZD1222 were shown to develop high levels of broadly neutralizing antibody responses following administration of a BNT162b2 booster [7, 13, 14]. This study also showed that the Sinopharm/BBIBP-CorV vaccinees who received the booster had high levels of ACE2 blocking antibodies (surrogate marker for neutralizing antibodies), although the positivity rates were lower than following natural infection. Indeed, similar observations have been made with vaccine recipients who received other vaccines, showing that infection acquired immunity boosted by a BNT162b2 vaccine, induced longer lasting immune responses. We found that while there was no difference to the antibody responses to the whole spike protein, S1 and S2 in cohort B, C and D, the antibody responses to N protein were highest in those who were infected prior to vaccination.

Although neutralizing antibodies to the spike protein are known to be associated with protection [14], the role of N protein specific antibody and T cell responses in protection is not well studied. Compared to the spike protein, the N protein is more conserved, and the SARS-CoV-2 variants have fewer mutations in the N protein than the S protein [25]. N protein has

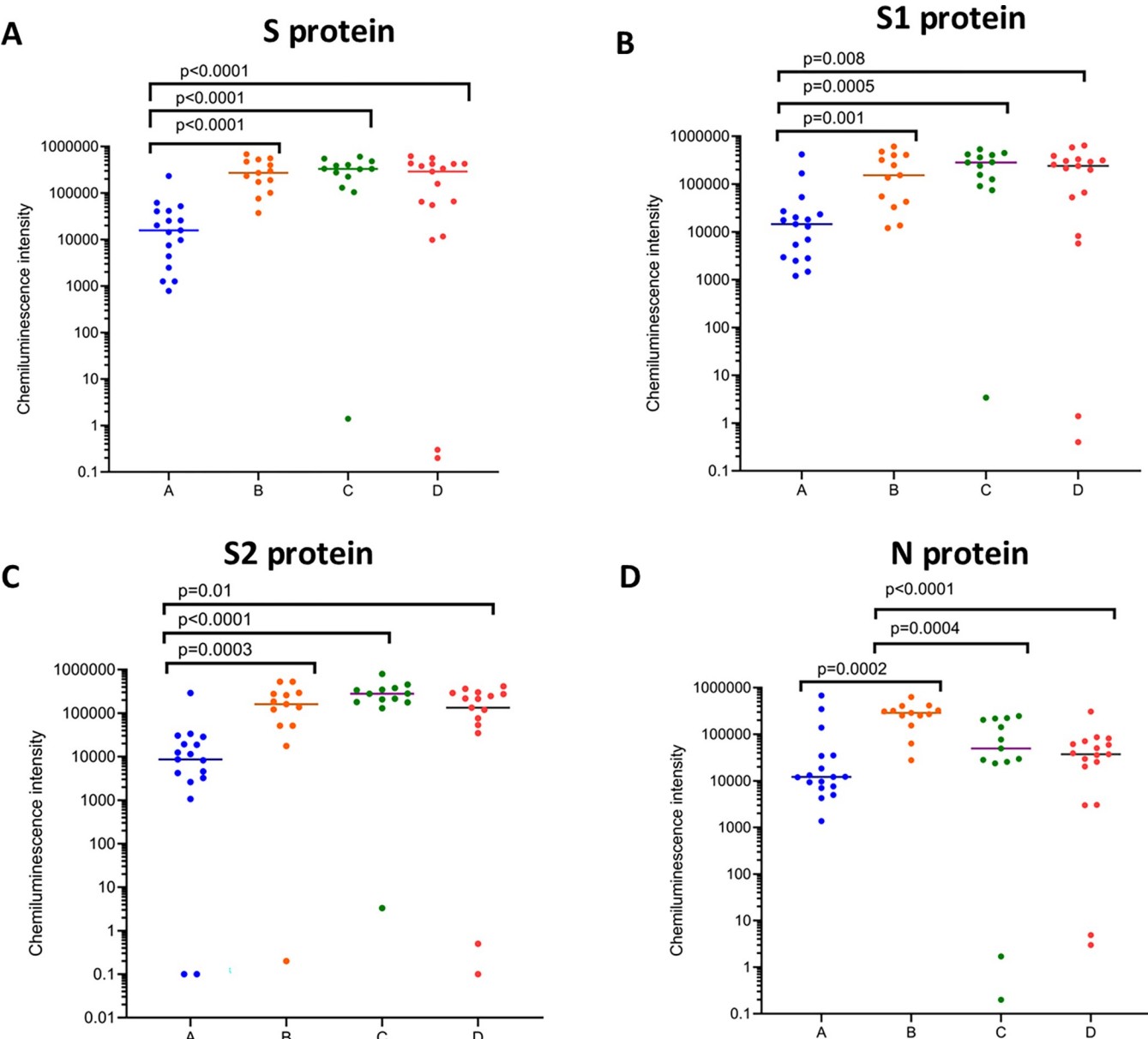

**Fig 3. Antibodies to the different SARS-COV-2 proteins in Sinopharm/ BBIBP-CorV vaccinees who either received the BNT162b2 booster or were infected.** IgG antibodies to SARS-CoV-2 spike (S), S1 subunit (S1), S2 subunit (S2) and nucleocapsid (N), were measured using the Peggy Sue platform, in Sinopharm vaccinees at 7 months post second dose, who remained uninfected and un-boosted (cohort A: n = 21), or those who had previous infection and who did not obtain the booster (cohort B: n = 17), or those who were not infected but who received a BNT162b2 booster (cohort C: n = 30), or those who did not receive the booster, but were infected with omicron (cohort D: n = 29). The Mann-Whitney test was used to compare the IgG levels between different cohorts for the different proteins. All tests were two-tailed.

shown to be an important antibody and T cell target [26]. Indeed, a high frequency of polyfunctional T cell responses specific for certain epitopes within the N protein was found to associate with milder illness [27]. Although we did not assess the T cell responses in this study, we show that those who had previous infection prior to receiving the Sinopharm/BBIBP-CorV vaccine, had the highest responses to the N protein compared to the other cohorts. Therefore, these data suggest that natural infection prior to vaccination may associate with a higher magnitude of immune responses, especially to the N protein.

Only 4.2% of Sri Lankans were fully vaccinated by 30th June 2021, when the delta variant rapidly spread in Sri Lanka [2]. Although seroprevalence data regarding the proportion of individuals who were naturally infected with the delta variant prior to vaccination is not known, by mid-July 2021 58/94 (61.7%) pregnant women recruited to another study, were found to be naturally infected prior to vaccination (Jeewandara et al, PLOS Global Health, Accepted). This indicates that many individuals in Sri Lanka could have been naturally infected with the SARS-CoV-2 virus prior to vaccination. Therefore, the reduced mortality rates in Sri Lanka compared to North America and Europe during rapid spread of BA.2, despite using less immunogenic vaccines and lower coverage with a BNT162b2 vaccine booster, could be due to a broader antibody and T cell response to the SARS-CoV-2, which should be further investigated.

## Conclusions

Those who were naturally infected with the SARS-CoV-2 virus prior to Sinopharm/BBIBP-CorV vaccine appear to have a high magnitude of antibody responses to different SARS-CoV-2 proteins and ACE2 receptor blocking antibodies, 7 months post vaccination. These individuals and those who did not receive the BNT162b2 vaccine booster, but were naturally infected with omicron, also had similar antibody responses to those who received the BNT162b2 vaccine booster. It will be important to investigate the implications for clinical protection in a larger cohort. In addition, immunity due to natural infection in individuals should also be considered, when planning vaccine booster programs.

## Supporting information

**S1 Fig. Recruitment and follow up plan of individuals who were previously infected or uninfected at the time of receiving the Sinopharm vaccine.**
(TIF)

**S2 Fig. Description of different cohorts who received the Sinopharm vaccine who either received or did not receive the BNT162b2 booster, or were infected with alpha, delta or the omicron variant.**
(TIF)

**S3 Fig. Automated Western Blot responses, assessed by the Peggy Sue platform to the five SARS-CoV-2 antigens in an individual who received the Sinopharm/BBIBP-CorV vaccine.**
(A-D) Graph images obtained from Compass SW. (E) Lane view in Compass SW. (A) at 6 weeks (B) at 3 months (C) at 6 months (D) Overlayed graph of A, B and C (E) Lane view of 6 weeks, 3 months, and 6 months of the same individual.
(TIF)

**S4 Fig.** The frequency (A) and phylogenic tree (B) of the SARS-CoV-2 variants identified in Sri Lanka from December 15th onwards. All sequencing data of our laboratory was uploaded to GISAID and the figures were obtained from Nextstrain [11].
(TIF)

## Author Contributions

**Conceptualization:** Chandima Jeewandara, Gathsaurie Neelika Malavige.

**Data curation:** Thashmi Nimasha, Jeewantha Jayamali, Maneshka Karunanada.

**Formal analysis:** Shashika Dayarathna, Thushali Ranasinghe, Gathsaurie Neelika Malavige.

**Funding acquisition:** Chandima Jeewandara, Graham S. Ogg, Gathsaurie Neelika Malavige.

**Investigation:** Inoka Sepali Aberathna, Shashika Dayarathna, Thashmi Nimasha, Thushali Ranasinghe, Lahiru Perera.

**Methodology:** Inoka Sepali Aberathna, Shashika Dayarathna.

**Project administration:** Chandima Jeewandara, Achala Kamaladasa.

**Resources:** Chandima Jeewandara, Graham S. Ogg.

**Supervision:** Chandima Jeewandara, Achala Kamaladasa, Gathsaurie Neelika Malavige.

**Writing – original draft:** Gathsaurie Neelika Malavige.

**Writing – review & editing:** Graham S. Ogg.

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
