## [Decision Letter · Decision Letter 0]

12 Aug 2022

PONE-D-22-17640Comparison of the kinetics and magnitude of antibody responses to different SARS-CoV-2 proteins in  Sinopharm/BBIBP-CorV vaccinees following the BNT162b2 booster or natural infectionPLOS ONE

Dear Dr. Malavige,

Thank you for submitting your manuscript to PLOS ONE. After careful consideration, we feel that it has merit but does not fully meet PLOS ONE’s publication criteria as it currently stands. Therefore, we invite you to submit a revised version of the manuscript that addresses the points raised during the review process.The reviewer has expressed serious concerns on the lack of sufficient demographic data, and details of correlating demographic details with antibody decay as well as a solid conclusion and hence a thorough attention on this regard may be warranted for further consideration. Thorough language editing is required as well.

We look forward to receiving your revised manuscript.

Kind regards,

Esaki M. Shankar, PhD, FRCPath

Academic Editor

PLOS ONE

Journal Requirements:

   "We are grateful to the funding by the Allergy Immunology and Cell Biology Unit of University of Sri Jayewardenepura; Sri Lanka Covid 19 Emergency Response and Health Systems Preparedness Project (ERHSP) of Ministry of Health Sri Lanka funded by World Bank and the UK Medical Research Council."

 "We are grateful to the funding by the Allergy Immunology and Cell Biology Unit of University of Sri Jayewardenepura (CJ); Sri Lanka Covid 19 Emergency Response and Health Systems Preparedness Project (ERHSP) of Ministry of Health Sri Lanka funded by World Bank (GNM) and the UK Medical Research Council (GSO). The funders had no role in study design, data collection and analysis, decision to publish, or preparation of the manuscript."

Reviewers' comments:

Reviewer's Responses to Questions

**Comments to the Author**

1. Is the manuscript technically sound, and do the data support the conclusions?

Reviewer #1: Partly

2. Has the statistical analysis been performed appropriately and rigorously? 

Reviewer #1: No

3. Have the authors made all data underlying the findings in their manuscript fully available?

Reviewer #1: No

4. Is the manuscript presented in an intelligible fashion and written in standard English?

Reviewer #1: No

5. Review Comments to the Author

Reviewer #1: Major comments

The title and conclusions in the abstract doesnot seem to match and hence the conclusions need to be rewritten. Abstract seem too lofty and needs to be abridged. The study is clearly in line with a similar study that suggests that natural antibodies are more durable although comorbidities and ageing seem to accelerate the antibody decay (doi: 10.3389/fmed.2022.887974 may be referred).

The study suffers from poor sample size and hence conclusions arrived appears to be unreliable.

A schematic illustration (algorithm) detailing on the cohort is needed to ease understanding of the different participant groups recruited into the investigation.

How was omicron infection detected in the vaccinees? How was SGTF overcome? No details available on the diagnosis of omicron. The authors must provide details of mutations (e.g. number of mutations, details of mutations, say for instance del69/70) and must provide a detailed phylogenetic analysis of the omicron variants that might provide clues to viral evolution kinetics in Sri Lanka.

The quality of immune responses to the vaccines tested is tantamount to, say for instance, expansion of antibodies that can neutralize the virus e.g anti-RBD antibodies. The authors did not evaluate the proportion of neutralizing antibodies but rather the total antibodies against the different epitopes of the virus. The authors must also clarify and define what are ACE2 antibodies. Does this mean the antibodies were specific to the ACE-2 (i.e. auto-antibodies to ACE-2)? If so, does it mean that vaccination can trigger anti-ACE2 antibodies?

Colclusion is not compelling and must translate the findings for better understanding of the take-home-message. It must advocate whether vaccination is required or if the pandemic evolution due to omicron has negated the need for further boosters.

Materials and methods:

Demographic characteristics, ethics approval, details of the vaccine lots used should be provided in detail. Type of study? The work schematics may be illustrated in a better way. The authors should have done some extensive investgation and statistical analysis to include age, comorbid conditions, and must have correlated with antibody longevity following vaccination and infection. This is a major limitation.

Minor comments

Line 6: Cohort C (not 'cohort c')

Line 62-6: Numbers/frequencies are dynamic and might change later.

Like 67: Refer doi: 10.3389/fmed.2022.887974, which also recommends a third dose given the rapid rate of decay of antibodies among the elderly as well as thse with underlying comorbid conditions.

Line 68: Statement pertinent to Africa might be misleading given the seemingly poor rates of reporting of both morbidity ad motrality and hence may be omitted, rather data available from South Asia and SE Asia may be relevant for mentioning herein.

Line 81: The date of first case of omicron in Sri Lanka worths a mention here.

Line: 84: Remove '(' bracket.

The authors must justify with cited references that a vaccine of poor immunogenicity was used in Sri Lanka.

Line 130: Infection status (not 'infectious status')

Line 214: ...a response... (edit)

Line 215: ....6 weeks...' -- previously mentioned as 2 weeks in abstract & materials & methods

Line 218: received received (edit)

6. PLOS authors have the option to publish the peer review history of their article (what does this mean?). If published, this will include your full peer review and any attached files.

Reviewer #1: No

---

## [Author Response · Author response to Decision Letter 0]

23 Aug 2022

21st August 2022

Esaki M. Shankar

Academic Editor

PLOS ONE

Dear Dr. Shankar,

Submission of a manuscript titled “Comparison of the kinetics and magnitude of antibody responses to different SARS-CoV-2 proteins in Sinopharm/BBIBP-CorV vaccinees following the BNT162b2 booster or natural infection"

We would like to thank the reviewer for carefully going through our manuscript and making useful suggestions. We have addressed all issues raised by the reviewer and have incorporated all these changes in the revised version. We have highlighted the changes in the revised.

Reviewer #1: Major Comments 

1. The title and conclusions in the abstract does not seem to match and hence the conclusions need to be rewritten. Abstract seem too lofty and needs to be abridged. The study is clearly in line with a similar study that suggests that natural antibodies are more durable although comorbidities and ageing seem to accelerate the antibody decay (doi: 10.3389/fmed.2022.887974 may be referred).

Response: We wish to thank the reviewer for these suggestions. We have amended the conclusions and the abstract. We thank the reviewer for sharing the reference, which shows the decay in the antibodies following different vaccines in those with comorbidities and in different ages. We have referred to this paper in the revised version. 

Our previous studies do show that the antibody responses decline faster to a greater extent in older individuals, for different vaccines, which assessed the responses the spike protein and also neutralizing antibodies (Jayathilaka et al, Clinical and Experimental Immunology, 2022; Jeewandara et al, Immunity Inflammation and Disease, 2022, June; Jeewandara et al, Immunity Inflammation and Disease, 2022, April). However, in this study, we sought to evaluate the kinetics and the magnitude of antibody responses to the different SARS-CoV-2 proteins, in naturally infected individuals, compared to those who received inactivated whole virus vaccines. This was also to assess the immunogenicity of the booster compared to natural omicron infection. 

2. The study suffers from poor sample size and hence conclusions arrived appears to be unreliable.

Response: We thank the reviewer for raising these concerns. In studies where the decay in antibodies is investigated in different age groups, we agree that larger sample sizes are required. We have used larger sample sizes in our previous studies (approximately 400 to 500), which investigated the decay and differences in the decay in antibody responses in different age groups and comorbidites. However, in this study we have assessed the differences in antibody responses to different proteins of the SARS-CoV-2 (S1, S2, N, S, RBS and ACE2 blocking antibodies) in different cohorts of individuals, who either received the Pfizer booster, or naturally infected with different SARS-CoV-2 variants. The aim of this study was to find out the differences in the breadth of the antibody responses in naturally infected individuals compared to those who received an inactivated whole virus vaccine and the immunogenicity of the booster vs natural infection. 

As the reviewer had raised concerns regarding the sample size, we have highlighted that the sample size is a limitation of our study. 

3. A schematic illustration (algorithm) detailing on the cohort is needed to ease understanding of the different participant groups recruited into the investigation.

How was omicron infection detected in the vaccinees? How was SGTF overcome? No details available on the diagnosis of omicron. The authors must provide details of mutations (e.g. number of mutations, details of mutations, say for instance del69/70) and must provide a detailed phylogenetic analysis of the omicron variants that might provide clues to viral evolution kinetics in Sri Lanka.

Response: We thank the reviewer for this important suggestion. We have included two figures (supplementary figure 1 and 2) describing the cohorts. 

Omicron infection was confirmed by whole genomic sequencing as described in the methods. All individuals included in this study as having an omicron infection were confirmed by sequencing, including those who received vaccines. All sequencing data are uploaded to Nextstrain and GISAID https://nextstrain.org/community/aicbu/ncov/srilanka. We have included a phylogenetic tree of the omicron sub-lineages present in Sri Lanka at this time (supplementary figure 4), which includes the sequencing data of all these individuals included in the study. Individuals were classified as having a BA.1, BA.1.1 and BA.2 infection, based on the whole genomic sequencing data. 

If the reviewer believes it is important to describe the mutations and the phylogenetic analysis of omicron variants in Sri Lanka, we are happy to do so and include a table and separate figures regarding these. Such an analysis and molecular epidemiology of the omicron sub lineages in Sri Lanka, would be a different study by itself. We seek editorial advice regarding this. 

4. The quality of immune responses to the vaccines tested is tantamount to, say for instance, expansion of antibodies that can neutralize the virus e.g anti-RBD antibodies. The authors did not evaluate the proportion of neutralizing antibodies but rather the total antibodies against the different epitopes of the virus. The authors must also clarify and define what are ACE2 antibodies. Does this mean the antibodies were specific to the ACE-2 (i.e. auto-antibodies to ACE-2)? If so, does it mean that vaccination can trigger anti-ACE2 antibodies?

Response: We thank the reviewer for this question. The ACE2 antibodies were measured by the surrogate virus neutralizing test, which is a surrogate measure of neutralizing antibodies. It has been used in many large epidemiological studies and vaccine studies to measure neutralizing antibodies. We have included relevant references in the revised version and give some references using this assay to measure neutralizing antibodies below.

1. Tan et al, A SARS-CoV-2 surrogate virus neutralization test based on antibody-mediated blockage of ACE2-spike protein-protein interaction. Nature Biotechnology, 2020

2. Chia et al, Dynamics of SARS-CoV-2 neutralising antibody responses and duration of immunity: a longitudinal study, The Lancet, 2021

3. Tan et al, Pan-Sarbecovirus Neutralizing Antibodies in BNT162b2-Immunized SARS-CoV-1 Survivors, NEJM, 2021

4. Jeewandara et al, Immune responses to a single dose of the AZD1222/Covishield vaccine in health care workers, Nature Communications, 2021

5. Conclusion is not compelling and must translate the findings for better understanding of the take-home-message. It must advocate whether vaccination is required or if the pandemic evolution due to omicron has negated the need for further boosters.

Response: We thank the reviewer for this comment. We have amended the conclusion accordingly. 

6. Demographic characteristics, ethics approval, details of the vaccine lots used should be provided in detail. Type of study? The work schematics may be illustrated in a better way. The authors should have done some extensive investigation and statistical analysis to include age, comorbid conditions, and must have correlated with antibody longevity following vaccination and infection. This is a major limitation.

Response: We are grateful for the above comment from the reviewer. We carried out this investigation as a prospective analytical study we have included ethics approval and vaccine lot details. Individuals who participated in the study were obtained vaccines from Sinopharm 2021020157 and 202105130807 batches. 

7. Minor comments

We thank the reviewer for drawing our attention to all the incidents mentioned here. We have corrected and incorporated all of them and are highlighted in our Revised manuscript with track changes. 

We appreciate the suggestions and comments made by the reviewer and we believe we could address all the concerns to the best of our ability. Thank you.

Yours Sincerely,

Prof. Neelika Malavige

---

## [Decision Letter · Decision Letter 1]

6 Sep 2022

Comparison of the kinetics and magnitude of antibody responses to different SARS-CoV-2 proteins in  Sinopharm/BBIBP-CorV vaccinees following the BNT162b2 booster or natural infection

PONE-D-22-17640R1

Dear Dr. Malavige,

We’re pleased to inform you that your manuscript has been judged scientifically suitable for publication and will be formally accepted for publication once it meets all outstanding technical requirements.

Kind regards,

Esaki M. Shankar, PhD, FRCPath

Academic Editor

PLOS ONE

Additional Editor Comments (optional):

Reviewers' comments:

Reviewer's Responses to Questions

**Comments to the Author**

1. If the authors have adequately addressed your comments raised in a previous round of review and you feel that this manuscript is now acceptable for publication, you may indicate that here to bypass the “Comments to the Author” section, enter your conflict of interest statement in the “Confidential to Editor” section, and submit your "Accept" recommendation.

Reviewer #1: All comments have been addressed

2. Is the manuscript technically sound, and do the data support the conclusions?

Reviewer #1: Yes

3. Has the statistical analysis been performed appropriately and rigorously? 

Reviewer #1: Yes

4. Have the authors made all data underlying the findings in their manuscript fully available?

Reviewer #1: Yes

5. Is the manuscript presented in an intelligible fashion and written in standard English?

Reviewer #1: Yes

6. Review Comments to the Author

Reviewer #1: (No Response)

7. PLOS authors have the option to publish the peer review history of their article (what does this mean?). If published, this will include your full peer review and any attached files.

Reviewer #1: No

---

## [Editor Report · Acceptance letter]

3 Oct 2022

PONE-D-22-17640R1 

Comparison of the kinetics and magnitude of antibody responses to different SARS-CoV-2 proteins in  Sinopharm/BBIBP-CorV vaccinees following the BNT162b2 booster or natural infection 

Dear Dr. Malavige:

I'm pleased to inform you that your manuscript has been deemed suitable for publication in PLOS ONE. Congratulations! Your manuscript is now with our production department. 

Kind regards, 

on behalf of

Dr. Esaki M. Shankar 

Academic Editor

PLOS ONE